# Acute psychological and physiological benefits of exercising with virtual reality

**Bradley Barbour**[1,2], **Lucy Sefton**[3], **Richard M. Bruce**[3], **Lucia Valmaggia**[1,4,5], **Oliver R. Runswick**[1]*

1 Department of Psychology, Institute of Psychiatry, Psychology & Neuroscience, King's College London, London, United Kingdom, 2 Institute of Health and Social Care School of Allied & Community Health London South Bank University, London, United Kingdom, 3 Centre for Applied Human and Physiological Sciences, Faculty of Life Science and Medicine, King's College London, London, United Kingdom, 4 Department of Psychiatry, KU Leuven, Leuven, Belgium, 5 Orygen, Centre for Youth Mental Health, The University of Melbourne, Parkville, Victoria, Australia

* oliver.runswick@kcl.ac.uk

**Data Availability Statement:** Data associated with this manuscript can be accessed via https://osf.io/km8tf/?view_only=d4cc902f72094eef831f9495cc6285bc.

## Abstract

Exercise is a powerful tool for disease prevention and rehabilitation. Commercially available virtual reality (VR) devices and apps offer an immersive platform to gamify exercise and potentially enhance physiological and psychological benefits. However, no work has compared immersive exercise to closely matched 2D screen-based equivalents with the same visual and auditory stimuli. This study aims to compare the acute effects of an exercise session using a commercial immersive VR workout to the same stimuli and workout presented on a screen. 17 healthy participants (male = 7, female = 10; aged 24.18±4.56 years), completed a 12-minute guided VR boxing exercise session in FitXR™ and a screen-based equivalent. Physiological responses were recorded continuously using a heart rate monitor and telemetricmetabolic cart system. Psychological and perceptual responses were measured using their ratings of perceived exertion, the physical activity enjoyment scale, and the physical activity affect scale. In the immersive VR participants chose to engage in more intense exercise (%VO2max; $p = 0.044$), showed higher levels of all enjoyment subscales ($p<0.05$) and reported higher positive affect ($p = 0.003$) and lower negative affect ($p = 0.045$) following exercise compared to the screen-based equivalent. However, the design here could not determine which elements of immersive VR contributed to the positive effects. Immersive VR may offer a more efficient alternative to other forms of screen based and exergaming workouts and could be offered as a gateway into exercise.

## Introduction

The positive effects of exercise and physical activity on health are well documented [1–3]. Most international governments recommend adults perform at least 150 minutes moderate intensity or 75 minutes vigorous intensity aerobic activity per week, with moderate intensity muscle strengthening activities performed two days per week [2, 4]. Despite the known benefits, over a third of the population remain physically inactive in England and almost half in

**Funding:** OR received contract research funding from FitXR https://fitxr.com/. The funders had no role in study design, data collection and analysis, decision to publish, or preparation of the manuscript.

**Competing interests:** OR received contract research fuding from FitXR

the USA [5, 6]. There is a significant need to develop strategies to support individuals in engaging with exercise and physical activity and technological developments have the potential to support this need. When testing new technological developments, it is important to include clear comparisons to existing options to establish if they offer further benefit.

Attrition rates for participating in new exercise programmes have been reported up to 58% [7], with 50% attrition to exercise programmes within 3–6 months [8]. Lack of time, motivation, facilities, social support, and other individual factors like anxiety and pain are amongst commonly reported perceived barriers to exercise [7, 9–11]. Poor education, exercise self-efficacy, and previous adverse experiences to formal exercise are also likely to decrease enjoyment and participation [12–15]. Researchers have investigated a myriad of interventions to enhance engagement with exercise and physical activity [16, 17]. A promising solution is exergames. These are digital games, originally played on 2D screens, that require physical activity to play and operate active gaming experiences [18] and have shown promise in supporting exercise in clinical populations such as those with chronic stroke symptoms [19].

Exergames have been available since the 1980s [20], however technological advances in the past decade have allowed higher fidelity and better interfaces during gaming including moving from 2D screens to immersive virtual reality (VR). VR involves a computer-generated environment that aims to induce a sense of being mentally or physically present in another place [21, 22]. This occurs on a spectrum depending on how visual information is displayed and the level of interaction available [23]. It could be two-dimensional, such as running with a screen-based environment displayed on a treadmill, or three-dimensional such as exercising in an environment rendered in a head mounted display [24]. Three-dimensional images provide depth and can be perceived as more immersive than two-dimensional, flat images which may offer semi-immersive experiences in contrast. With the most immersive forms of VR the user can physically interact with simulations, where information is passed from the user to the digital environment using the movement of the user, the environment can then be manipulated, such as striking a ball and seeing it fly [25]. The element of environmental interaction significantly influences perceptions of immersion and effort [26, 27] and emotional responses to exercise [28]. Immersive VR offers an opportunity for gamified exercise that could support retention in exercise programmes beyond what can be currently achieved in exergames using 2D screens. While barriers remain in the perception of cost [29] and potential for adverse responses like dizziness, standalone VR headsets like the one used in this study are available at lower cost than traditional games consols and have potential for enhanced effects. Research is required to assess the efficacy of these more immersive options with direct comparisons to more established technologies.

Efforts to apply new technologies to increase sustained exercise participation can utilise the COM-B model of behaviour change. This model suggests that behaviour (B) is dependent on three components: capability (C), opportunity (O) and motivation (M) [30–32]. VR can target all elements of the COM-B model. For example, the capabilities of individuals to exercise can be enhanced using VR. This is because, be it due to distraction [33, 34] or deliberate manipulation [27, 35], young and healthy participants can exercise at lower perceived exertion for the same workloads or choose to exercise for greater durations or at higher intensities when using VR [27, 34, 36, 37]. Motivation to exercise has also been shown to increase with the use of exergames [18] and compared to non-VR exergames, VR users report sessions to be more enjoyable [26, 38]. For example, Bird et al. [28] used the embodiment-presence-interactivity cube to conceptualise how increases in presence and interactivity in VR could impact the exercise experience. Results showed that interactive VR increases affective valence during exercise and remembered pleasure of exercise compared to less interactive equivalents. Such

psychosocial factors are predictive to engaging in physical activity and exercise [39, 40] and the convenient and accessible nature of VR offers opportunities for fun social interaction via multiplayer functions and competitions without needing to leave the house. VR could, therefore, enhance a person's ability to reach physical activity guidelines and achieve the related benefits for physical and psychological outcomes [41–44]. In clinical populations, greater improvements in function and quality of life have been reported in those living with neurological conditions compared to conventional approaches [45, 46]. However, some participants may prefer conventional exercise for greater social interaction and the effectiveness may be dependent on user acceptance of technology [46].

Despite the possible benefits of gamifying exercise and the potential additional benefit to chosen work rate, researchers have often artificially controlled the participants' exercise workload or compared immersive VR to rest, rather than equivalent non-immersive or semi-immersive exercise. No work in healthy individuals or clinical populations has used immersive VR exercise and a closely matched 2D equivalent while capturing gold standard measures of physical workload and allowing participants autonomy to exercise at a chosen intensity. Therefore, this study aims to use a commercially available device and app to investigate the acute physiological and psychological effects of an immersive VR workout compared to a screen-based workout where the visual and auditory stimuli are matched. This will be achieved through taking measures of physiological workload (Heart rate, HR, oxygen consumption), perceptions of effort, enjoyment and affect, as well as the acceptability, feasibility, and tolerability of using VR for exercise.

Participants completed $VO_2$ max testing and engaged in the same boxing workout both in an interactive and immersive VR setting or with the exact same stimuli visual and auditory stimuli appearing on a large 2D screen. We hypothesised that in the immersive 'VR' condition participants will choose to exercise at a higher intensity and this will produce a greater physiological response through higher heart rates and $VO_2$ across the whole workout compared to the 'screen' condition. This will be linked to a lower perceived amount of effort, measured through RPE. We also hypothesised that participants would enjoy the immersive VR exercise more and report higher scores on the physical activity enjoyment scale on the positive affect dimension of the physical activity affect scale [47].

## Method

### Participants

We performed a sample size calculation using G*Power (3.1.9.7) based upon the RPE effect sizes from Zeng et al. [34] who recruited the same student population and measured within subject differences in RPE between immersive VR, screen-based semi-immersive VR, and a traditional cycling condition (dz = 0.85). Using a matched pairs t-test to represent hypothesised between condition differences, an α of 0.05, and Power of 0.95 (selected to historic issues with low power in this field), we calculated a required sample size of 17. A convenience sample of 17 healthy individuals from the student population responded to on campus adverts and volunteered to take part (aged = 24.2±4.6 years; height = 168.21±10.49cm; body mass = 69.62 ±14.43kg; BMI24.47±4.04kg/$M^2$). Participants were required to be novices to VR and individuals who trained less than 5 days/week, those who reported limitations to physical exertion and exercise capacity such as medical conditions (e.g., COPD, CVD) or any history of metabolic or respiratory disease were excluded. Participation was entirely voluntary with no financial incentive. This study was approved by the local Research Ethics Committee (MRPP-22/23/3691) and all participants provided informed consent.

## Materials and stimuli

Immersive VR exercise was conducted using FitXR™ on a Meta Quest 2 VR headset with two handheld controllers. A polar strapped HR monitor (Polar H10) was attached with contact with participant's sternum. $VO_2$ was measured using a calibrated telemetric metabolic cart system (Metamax 3B), attached to an oronasal mask which participants wore throughout the activity (Fig 1). Prior to use of the headset, participants received verbal instructions for controller use and navigating the virtual and physical environment safely. Participants were then required to navigate the menus themselves to show proficiency in the use of the headset and app. Once in the app participants navigated to an intermediate difficulty boxing workout called 'pack a bunch' with personal trainer Dillon supplying the instructional voiceover. The boxing workout consists of orbs flying towards participants with a white light glowing from one side to indicate the type of punch required. Each workout also incorporates ducks, weaves, and blocks. The activity duration was 12 minutes 25 seconds. The music and a coach verbally encouraged and guided users through four consecutive rounds that are the default option in the game (warm-up, defense & counter, conditioning, and fight; see Fig 2). The four round element allows for responses to different intensities of exercise stimuli to be observed. FitXR™ allows users to select a location, all participants conducted this workout in the 'rooftop' setting. For the screen-based equivalent we created a 2D screen-based stimuli by screen recording the same workout from the headset meaning the two workouts out were identical aside from the immersive nature of the VR version and the visuo-haptic feedback and real-time scores were visible to users in VR. The experimental set up was the same as the immersive VR condition in Fig 1, but without the head mounted display. No feedback or scores were recorded for the screen condition, given the nature of the task and technology.

## Protocol

All participants performed the activity using an immersive 'VR' and 'Screen' condition. The order of conditions was counterbalanced. Both sessions began with a 1-minute rest period, where baseline physiological measures were recorded. There was an approximate 5-minute rest period between sessions to allow participants to return to their resting heart rate. Participants HR was measured while standing at rest for one-minute before each workout and a paired samples t-test showed no significant difference between resting HR prior to starting each condition (VR = 84±18, Screen = 86±15, $t = 0.814$, $p = 0.428$, $d = 0.20$). After both versions were completed, participants undertook a Modified Balke Protocol walking treadmill $VO_2$ max test [48]. A 1-minute rest period preceded walking to obtain baseline measures. Due to sex-specific differences, the initial walking speed was gender dependent (males = 3.3 mph, females = 3.0mph). The incline was increased by 2% after 2 minutes of walking and by a further 2% every proceeding minute. If a maximum incline of 20% was achieved, speed would increase by 0.2mph every minute. Participants were instructed to continue up to maximal effort. Maximal $VO_2$ ($VO_2$ max) was considered to be achieved if HR reached +/-10% of the predicted maximal value (220bpm–age). All participants achieved this criterion.

## Measures

**Physiological measures.**   Throughout the workouts we collected heart rate (HR) in beats per minute (bpm) and oxygen consumption ($VO_2$) in ml/kgmin. Maximal oxygen ($VO_2$ max) was calculated as the highest mean $VO_2$ over a 20 second period and percentage of $VO_2$ max during activities were calculated. Measures were collected across all stages of the workout to establish if effects diminished over time due to the novelty present at the start.

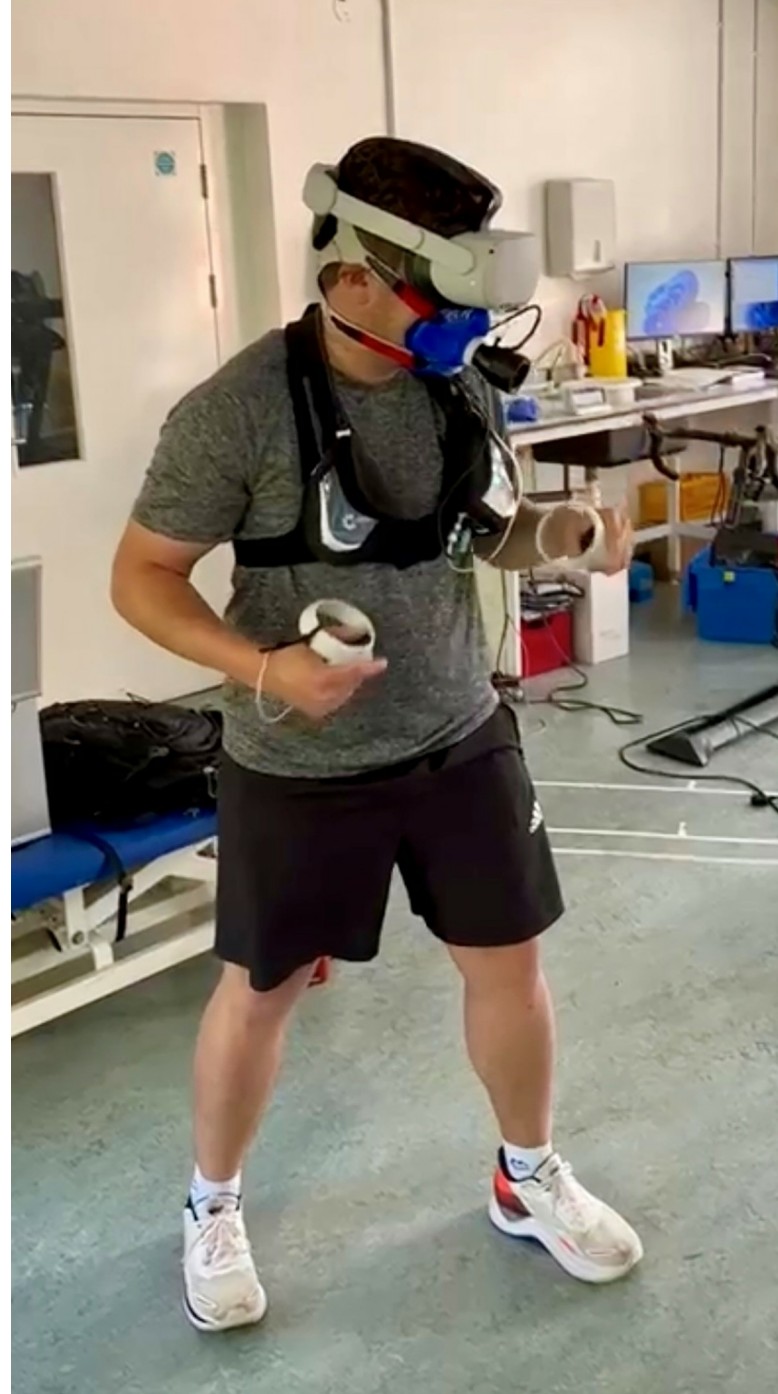

**Fig 1. Participant wearing the headset and portable metabolic cart.**

**Exercise perception and affective responses.** Incrementally, rate of perceived exertion (RPE) scores was reported by participants verbally using the CR10 Borg scale [49], which participants were familiarized with before putting on the VR headset. The modified 1–10 scale was the preferred scoring system for simplicity and to allow intuitive reporting of RPE, particularly if participants were unfamiliar with the Borg's original 6–20 scale [50]. Three short

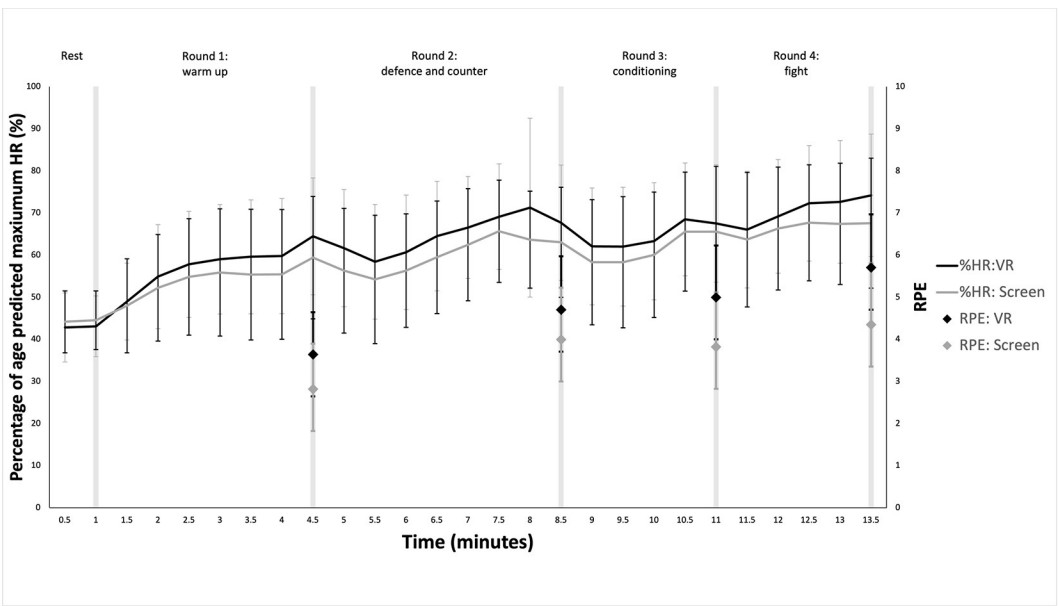

**Fig 2. %HRmax and RPE at 30 second intervals across the VR and screen workouts.** Error bars represent standard deviations.

Likert style self-assessment questionnaires were administered upon completion of each condition. The Physical Activity Enjoyment Scale (PACES) questionnaire was administered to quantify enjoyment of activity [51]. A five-point Likert style Physical Activity Affect Scale (PAAS) questionnaire distinguished the psychological response to each condition, ranging from '1—do not feel' to '5—feel very strongly' [52]. Subscales of the PAAS are used to categorise impacts as positive (upbeat, energetic and enthusiastic), negative (miserable, discouraged and crummy), tranquility (calm, peaceful, relaxed) or fatigue (tired, worn-out and fatigued), as demonstrated by Lox et al. [52]. The post-VR questionnaires proceeded with subjective insights regarding motivations for using VR, engagement, fears and barriers to using VR.

**Feasibility, acceptability, and tolerability.**   Participants were asked to answer a series of questions after the VR condition, designed to gather participant feedback of the acceptability, tolerability, and feasibility of exercising in virtual reality. Simulation Sickness Questionnaire (SSQ) [53] to assess tolerability and potential cybersickness effects and the open-ended questions were asked to further investigate acceptability and feasibility (see Table 1).

**Table 1. Follow up questions that participants were asked after completing the VR exercise workout and the reason for these questions based on Birckhead et al. [54].**

| Question/Measure | Category |
|---|---|
| Simulation Sickness Questionnaire | Tolerability |
| What motivated you to participate in this virtual reality and exercise-based research? | Acceptability |
| Did you have any worries or fears about using virtual reality during exercise? If yes, please tell us what they were. | Feasibility |
| What do you think using virtual reality can add to your exercise sessions? | Feasibility |
| Would you like to use virtual reality again in an exercise session at another time and why? | Acceptability |
| How often would you like to use virtual reality? It could from any time you exercise, every once in a while, or never! | Feasibility |
| Are then any barriers that would prevent you from using virtual reality during exercise? | Acceptability/ Tolerability |

## Data analysis

Breath-by-breath $VO_2$ and HR data was recorded and extracted for the MetaSoft studio software at 30 second intervals. JASP 0.18 was used for statistical analyses. The mean percentage of age predicted maximal HR (%HRmax) and $VO_2$ for each 30 second epoch in each round were calculated to create a round score for warm-up, defense & counter, conditioning, and fight. To measure effects of condition and round on RPE, HR, and $VO_2$ we used a 2 condition × four time point repeated measures ANOVA. A Bonferroni adjustment was employed when multiple comparisons were being made for time points in order to lower the significance threshold and avoid Type I errors. Violations of sphericity were corrected by adjusting the degrees of freedom using the Greenhouse–Geisser correction when ε was less than 0.75 and the Huynh–Feldt correction when greater than 0.75. Partial eta squared was used as a measure of effect size for all analyses. To compare enjoyment and affective responses in Immersive VR compared to Screen we used paired sample t-tests for each subscale. All comparisons made were pre-planned; therefore, alpha value was kept at *p = .05* and effect sizes (Cohen's d) and 95% confidence intervals were reported [55].

We also collected feedback to explore opportunities to exercise in VR focusing on participants perceptions of acceptability, feasibility, and tolerability [54]. For yes or no answers we simply present descriptive counts. Where participants expanded on answers, we were not able to generate categories for a content analysis based on previous literature due to the lack of previous work in the area. Therefore, to analyse the follow-up questions, we used a blended approach where we first analysed interview content inductively to produce themes that could then be used for categories in a count-based content analysis [56]. This process was conducted for each question and participants were not limited to single codes per question.

## Results

### Physiological responses

**Heart rate.** Fig 2 shows how %HRmax changed throughout each round in both conditions. There was a main effect of time on %HRmax ($p < 0.001$, F = 71.898, $\eta^2 = 0.681$). These differences occurred between '*Rest*' and '*Round 1*' (mean difference = -12.46, p = <0.001, d = -0.98, 95%CI = -17.10 - -7.81), '*Round 1*' and '*Round 2*' (mean difference = -6.45, p = 0.001, d = -0.51, 95%CI = -11.097 - -1.80) and '*Round 3*' and '*Round 4*' (mean difference = -5.59, p = 0.009, d = -0.04, 95%CI = 10.24 - -0.94) but not between '*Round 2*' and '*Round 3*' (mean difference = -0.56, p = 1.000, d = -0.04, 95%CI = -5.21–4.09). Mean±SD %HRmax values (%) during each round were: '*Rest*' VR = 42.96±7.09, Screen = 44.35±7.60; '*Round 1*' VR = 57.78 ±13.56, Screen = 54.43±12.45; '*Round 2*' VR = 64.97±13.03, Screen = 60.14±13.23; '*Round 3*' VR = 64.67±14.76, Screen = 61.56±13.49; '*Round 4*' VR = 70.90±29, Screen = 66.53±13.95. There was no significant effect between exercise type (p = 0.064, F = 3.957, $\eta^2 = 0.019$) and no significant time-type interaction (p = 0.079 (G-G), F = 2.991, $\eta^2 = 0.011$). Peak %HRmax for VR was 74.19% and 67.67% for Screen. The mean difference between exercise types was 2.85%.

**$VO_2$.** There was a significant main effect of time (p = <0.001, F = 68.853, $\eta^2 = 0.695$) and type of exercise (p = 0.044, F = 4.765, $\eta^2 = 0.019$) on $VO_2$. There was no significant time-type interaction (p = 0.289 (G-G), F = 1.281, $\eta^2 = 0.004$). Fig 3 shows how $VO_2$ (ml/min/kg) changed throughout both exercise types. Mean±SD VO2 during each round were: '*Rest*' VR = 5.31±1.23, Screen = 5.03±0.99; '*Round 1*' VR = 14.75±7.24, Screen = 12.87±6.32; '*Round 2*' VR = 20.00±5.64, Screen = 17.56±7.11; '*Round 3*' VR = 18.33±6.80, Screen = 16.57±7.48; '*Round 4*' VR = 22.43±8.39, Screen = 19.56±8.29. Post Hoc comparisons showed differences

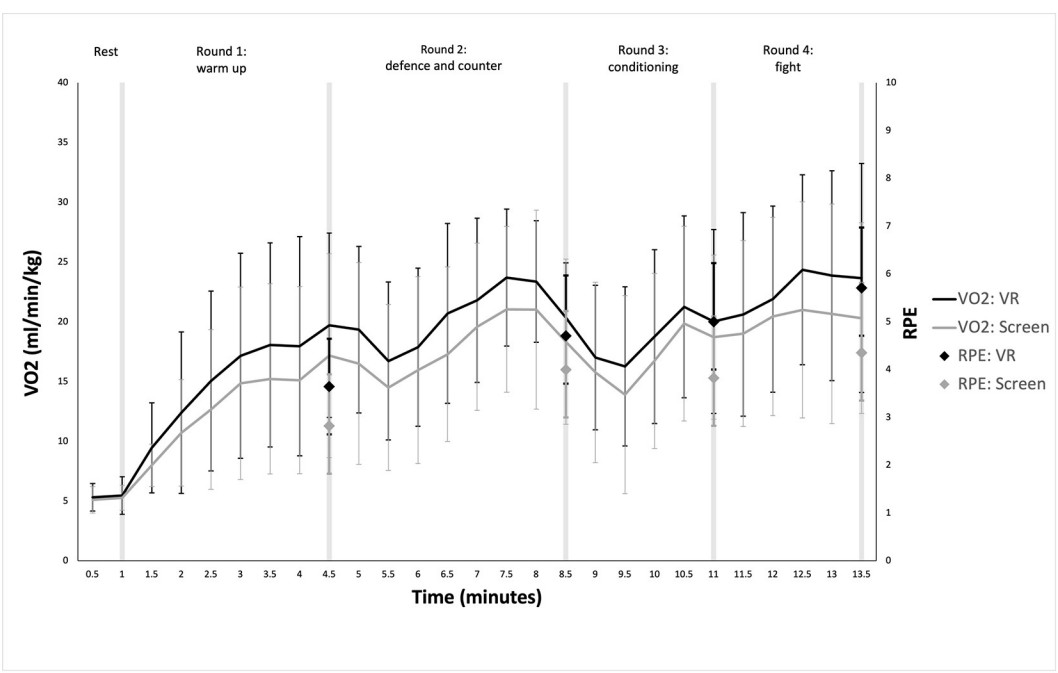

**Fig 3. VO2 and RPE at 30 second intervals across the VR and screen workouts.** Error bars represent standard deviations.

occurred for time between '*Rest*' and '*Round 1*' (mean difference = -8.64, p = <0.001, d = -1.34, 95%CI = -11.72 - -5.57), '*Round 1*' and '*Round 2*' (mean difference = -4.97, p = <0.001, d = -0.77, 95%CI = -8.05 - -1.87) and '*Round 3*' and '*Round 4*' (mean difference = -3.54,

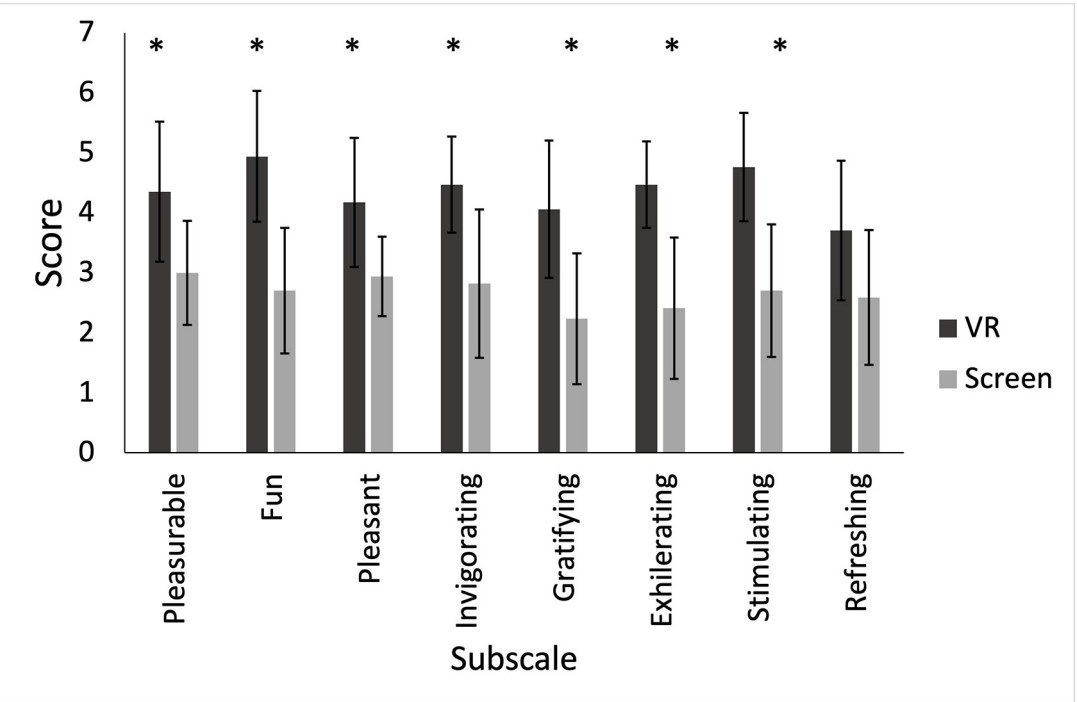

**Fig 4. PACES subscales for the VR and screen workouts.** * Denotes significant differences.

p = 0.014, d = -0.55, 95%CI = -6.62 - -0.47), but not between 'Round 2' and 'Round 3' (mean difference = 1.3, p = 1.00, d = 0.21, 95%CI = -1.74–4.41). Post Hoc comparison of type was significant (mean difference = 1.85, p = 0.044, d = 0.29 95%CI = 0.05–3.64). The average $VO_2max$ (ml/kg/min) for participants was 45.41±9.52.

## Psychological responses

**Rate of perceived exertion.** Mean RPE values for VR and Screen at the end of each round were: 'Round 1' = 3.6±1.0 and 2.8±1.1, 'Round 2' = 4.7±1.3 and 4.0±1.2, 'Round 3' = 5.0±1.2 and 3.8±1.3, 'Round' 4 = 5.7±1.3 and 4.3±1.5. There was main effect of both time (p<0.001, F = 51.379, $\eta^2$ = 0.403) and exercise type (p<0.001, F = 45.289, $\eta^2$ = 0.248). For time, these differences existed between 'Round 1' and 'Round 2' (mean difference = -1.12, p<0.001, d = -0.91, 95%CI = -1.50 - -0.32), 'Round 3' and 'Round 4' (mean difference = -0.62, p<0.001, d = -0.50, 95%CI = -0.94 - -0.07), but not between 'Round 2' and 'Round 3' (mean difference = -0.06, p = 0.100, d = -0.05, 95%CI = -0.40–0.30). Post hoc comparisons for type were significant (mean difference = 1.02, p<0.001, d = 0.83, 95%CI = 0.43–1.22). The was no significant time-type interaction (p = 0.100, F = 2.205, $\eta^2$ = 0.016).

**Exercise perception and affective responses.** Fig 4 shows PACES results. Total scores show immersive VR performed superiorly to Screen (VR = 4.4±0.9, Screen = 2.7±0.8, p<0.001, d = 1.291, 95%CI = 0.630–1.930). VR exercise performed greater than Screen exercise for subscales; pleasurable (VR = 4.4±1.2, Screen:3.0±0.2, p<0.001, d = 1.064, 95%CI = 0.454–1.653), fun (VR = 4.9±1.1, Screen = 2.7±1.0, p<0.001, d = 1.363, 95%CI = 0.685–2.019), pleasant (VR = 4.2±1.1, Screen = 2.9±0.7, p<0.001, d = 1.132, 95%CI = 0.507–1.735), invigorating (VR = 4.5±0.8, Screen = 2.8±1.2, p<0.001, d = 0.995, 95%CI = 0.399–1.570), gratifying (VR = 4.1±1.1, Screen = 2.2±1.1, p<0.001, d = 1.070, 95%CI = 0.459–1.660), exhilarating (VR = 4.5±0.7, Screen = 2.4±1.2, p<0.001, d = 1.4435, 95%CI = 0.739–2.109) and stimulating (VR = 4.8±0.9, Screen = 2.7±1.1, p<0.001, d = 1.355, 95%CI = 0.679–2.009). No differences

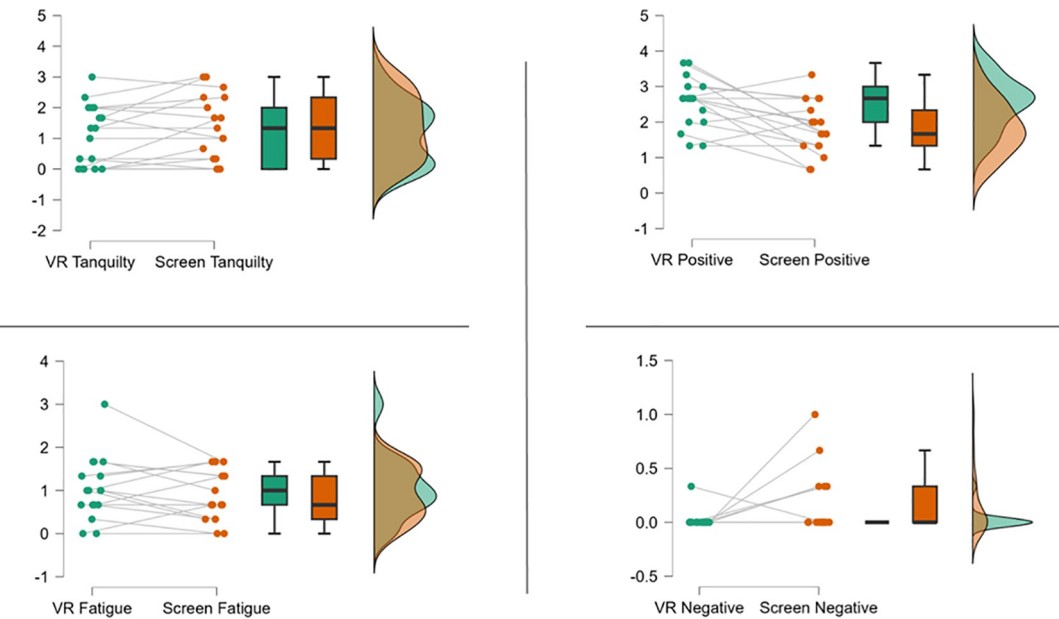

**Fig 5.** PAAS results showing individual scores, means, and distributions for positive affect (top left), tranquility (top right), negative affect (bottom left) and fatigue (bottom right).

were detected in refreshing subscale (VR = 3.5±1.125, Screen = 2.6±1.1, p = 0.051, d = 0.511, 95%CI = -0.003–1.010).

Fig 5 presents the outcomes of PAAS. There was greater positive affect post-VR exercise (VR = 2.5±0.7, Screen = 1.8±0.7, *p* = 0.003, d = 0.774). There were no significant differences in tranquility (VR = 1.1±1.0, Screen = 1.4±1.0, p = 0.943, d = -0.405). Compared to Screen, there was less negative affect in VR (VR = 0.02±0.1, Screen = 0.2±0.3, p = 0.045, d = -0.438). There were no differences in fatigue (VR = 1.0±0.7, Screen = 0.9±0.6, p = 0.839, d = 0.248).

## Feasibility, acceptability, and tolerability

No participants reported feeling motion sick after each condition. SSQ results indicated that general discomfort was reported in VR compared to Screen (VR = 0.5±0.8, Screen = 0.1±0.3, p = 0.014, d = 0.666). No significant differences were found in all other subscales (p>0.05). Responses to questions on feasibility and acceptability can be found in Table 2. Almost all participants saw VR exercise as a feasible and acceptability addition to their exercise programmes. This generally focused on an ability to add variety to exercise, increase enjoyment and add competition and motivation. 15 out of the 17 respondents would use VR again for exercise and the majority suggested that this would be on a regular basis or once a week or more.

## Discussion

This study aimed to use a commercially available device and app to investigate the acute physiological and psychological effects of immersive VR exercise compared to exercising with the same visual and auditory stimuli presented on a 2D-screen. We took measures of workload (% HRmax and $VO_2$), perceptions of effort, enjoyment and affect, as well as the acceptability, feasibility, and tolerability of using immersive VR for exercise. Results showed that participants chose to engage in work that consumed approximately three ml/kg/min of oxygen more in immersive VR compared to a screen-based equivalent, but with no differences in perceived exertion. The VR exercise was also rated as more enjoyable and resulted in more positive and less negative affective responses. VR exercise was tolerable and was perceived as feasible and acceptable by the participants.

As hypothesized, $VO_2$ values were higher during VR exercise compared to Screen, % HRmax did not show a significant main effect of exercise type but did show a small effect. However, contrary to our predictions, participants did also perceive these high levels of exertion, evidenced in higher RPE scores in VR. Findings partially support those of Runswick et al. [27] who found participants cycled at higher intensities in VR but did not perceive the exertion to be higher. Similar findings were reported by Glen et al. [57] who found greater RPE values during exergame conditions compared to a blank screen. The selection of higher workload was selected that is also perceived, this does not support a wider body of work that has shown reductions in perceptions of exercise intensity [34, 58] or breathlessness [35] in virtual environments. In this study we aimed to allow participants to exercise as they would naturally in VR, rather than control workload, as many studies focusing on reducing RPE have done. The higher work rate in VR may have been influenced by factors, such as enjoyment and live feedback and haptics from the controllers.

As anticipated, enjoyment and positive affect scores were higher for VR exercise, despite the higher work rates. In traditional exercise settings, enjoyment can decrease as intensity increases [59], however, here we have shown higher levels of exercise intensity and enjoyment, suggesting a benefit from exercising in a gamified way. Our findings are consistent with several studies that find VR to be enjoyable [33, 60]. The environmental interactivity of VR exercise may have driven this affect where haptic feedback can positively influence the experience of

**Table 2. Themes and example quote alongside content counts or each follow-up question.**

| Question | Theme | Example | Count |
|---|---|---|---|
| What motivated you to participate in this virtual reality and exercise-based research? | Interest in VR | "I wanted to try VR" | 8 |
| | Novelty | "I wanted to try something new" | 4 |
| | Interest in Research | "The ball didn't bounce accurately" | 3 |
| | Fun | "A bit of fun to experience a VR workout" | 3 |
| Did you have any worries or fears about using virtual reality during exercise? | No worries or fears | "None" | 11 |
| | Spatial awareness | "Bumping into things" | 3 |
| | Self-consciousness | "Looking stupid" | 1 |
| | Sickness | "Slightly worried about motion sickness" | 1 |
| | Equipment | "Wearing the equipment" | 1 |
| What do you think using virtual reality can add to your exercise sessions? | Fun | "Makes it more fun and creative way to exercise" | 6 |
| | Variety | "Offers variety of different activities that I wouldn't do normally" | 6 |
| | Motivation | "Definitely adds motivation when it feels real" | 5 |
| | Feedback/ competition | "Really enjoy the competitiveness, points system—compete against self or others" | 4 |
| | Accessibility | "Good alternative if you can't go outside/ don't have access to facilities" | 2 |
| | Engagement | "It adds engagement and distraction" | 1 |
| Would you like to use virtual reality again in an exercise session at another time and why? | Yes—Fun | "Yes, because it was fun" | 7 |
| | Yes—Motivation | "Yes, because it's more engaging and motivating" | 3 |
| | Yes–Exercise perceptions | "Yes, it was more fun and didn't feel like 'exercise'" | 2 |
| | Yes–Variety | "Yes, because it's different to conventional exercise" | 2 |
| | Yes–Schedule | "Yes, easy to fit in with schedule" | 1 |
| | No—Ergonomics | "No because it was clunky, didn't allow proper movement" | 1 |
| | No—Space | "No because of lack of space and money" | 1 |
| | No—Cost | "No because of lack of space and money" | 1 |
| How often would you like to use virtual reality for exercise? | Once a week or more | "Once or twice a week" | 11 |
| | Every day | "Daily" | 2 |
| | Once a month or more | "Maybe twice a month" | 3 |
| | Never | "Never" | 1 |
| Are then any barriers that would prevent you from using virtual reality during exercise? | Cost* | "Price accessibility" | 8 |
| | Space | "Space" | 6 |
| | Equipment | "Technical difficulty of set-up" | 3 |
| | No | "No | 4 |

*Participants were not informed how much headsets or app cost.

VR users and improve performance [61–63]. Simulating kinesthetic information such as force and pressure with punches in VR via haptic feedback, may have increase sensory fidelity, sense of presence and immersion by engaging senses beyond the visuo-audio of Screen [64, 65]. Adding to this previous work and supporting that of Teixeira et al. [44], we also captured the increase in positive emotions and decrease in negative emotions after the workout.

As well as recording higher workloads, more enjoyment and more positive affect when exercising visiting the lab, we also aimed to collect information on how the participants perceived the feasibility, acceptability, and tolerability of using VR in the future [44]. No participants experienced any adverse effects of using the HMD, suggesting that the exercise is

tolerable. Content analysis of the subjective responses showed that exercise in VR was likely acceptable and feasible for these participants as well. The majority reported a desire to use VR for exercise on a regular basis due to increases in motivation and enjoyment and that VR exercise had few perceived barriers.

Revisiting the COM-B model of behaviour change, our findings suggest that VR could have potential to impact exercise behaviour through increases in capability (C), opportunity (O) and motivation (M) [30–32]. The immersive VR condition here supported participants in choosing slightly more intense exercise as evidence with increased oxygen uptake, suggesting the capability to engage in higher intensity exercise. The increases in positive affect and enjoyment have potential to add motivation, as was supported by the feedback from participants. Improved enjoyment can enhance self-reported self-efficacy for exercise and attitudes towards exercise [66]. Participants also suggested that VR could increase the opportunity to engage in more regular, higher intensity, and more enjoyable exercise. It is well established that gamification of exercise can improve enjoyment and attitude towards exercise, as well as shaping behaviour to increase exercise activity [67, 68] but the findings here suggest that VR has potential to build on this further. Future work is needed to see whether these acute effects can indeed predict behavior change over the longer term.

## Limitations

Whilst these results are promising, they should be considered in light of the limitations of this study and its design. Firstly, novelty may play a role in the findings reported here from acute bouts of exercise. Participants were new to VR, and many reported they took part due to the novelty and were interested in trying VR (Table 2). This may lead to higher levels of enjoyment and effort in the VR condition and future research should investigate the use of VR over longer training periods, considering how these effects compare over time with diminishing novelty. In our design using matched visual and auditory stimuli on a 2D screen and the fully interactive VR condition, from this we cannot determine whether it was the 360 immersion, live feedback, haptics from controllers or other factors that led underpinned the differences displayed. Additionally, our cohort was relatively young and active. Therefore, our findings may not be applicable to the wider general population or older aged adults. Future research should focus on these populations, where it is possible positive impacts could be greater for those who do not engage in exercise at all.

In this study, a common comment by participants was that the VR headset could not be worn perfectly flush over the mid portion of the nose with the ventilatory mask on. This led to some discomfort and potentially had some effect on display clarity and adding to the sweating caused the VR headset. When wearing the VR headset, participants could not see Borg scale and did so verbally after being familiarized with the visual scale beforehand. We used the CR10 version to make this easier, but participants were required to report RPE using their memory of scale points. A priori measure of acceptability and feasibility were not characterized, nor were baseline SSQ measures obtained. There were some large individual differences in enjoyment and affective responses to VR in this study. It is possible that some individuals less engaged or interested in VR may be 'non-responders'. While we do not have enough power here for group-based analysis, future research may benefit from analysing individuals who do and do not enjoy VR.

## Conclusion

Exercise in VR can increase chosen work rate, activity enjoyment, and elicit positive psychological responses compared to a non-VR equivalent in young healthy participants. Health

promoting efforts may be enhanced by suggesting VR as an alternative form of exercise. Whilst VR use continues to expand, future research should investigate the implementation of such technologies into other populations, in healthcare systems, and establish whether they can be beneficial for enhancing disease prevention and condition management over long term interventions.

## Acknowledgments

The authors would like to thank Brendon Stubbs for the support with this project.

## Author Contributions

**Conceptualization:** Lucy Sefton, Richard M. Bruce, Lucia Valmaggia, Oliver R. Runswick.

**Data curation:** Bradley Barbour, Lucy Sefton, Oliver R. Runswick.

**Formal analysis:** Bradley Barbour, Oliver R. Runswick.

**Funding acquisition:** Lucia Valmaggia, Oliver R. Runswick.

**Investigation:** Richard M. Bruce, Oliver R. Runswick.

**Methodology:** Lucy Sefton, Richard M. Bruce, Lucia Valmaggia, Oliver R. Runswick.

**Project administration:** Lucy Sefton, Oliver R. Runswick.

**Resources:** Richard M. Bruce.

**Supervision:** Richard M. Bruce, Oliver R. Runswick.

**Visualization:** Oliver R. Runswick.

**Writing – original draft:** Bradley Barbour, Oliver R. Runswick.

**Writing – review & editing:** Bradley Barbour, Lucy Sefton, Richard M. Bruce, Lucia Valmaggia, Oliver R. Runswick.

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
