## [Decision Letter · Decision Letter 0]

22 Jul 2024

PONE-D-24-05252Acute psychological and physiological benefits of exercising with virtual realityPLOS ONE

Dear Dr. Runswick,

Thank you for submitting your manuscript to PLOS ONE. After careful consideration, we feel that it has merit but does not fully meet PLOS ONE’s publication criteria as it currently stands. Therefore, we invite you to submit a revised version of the manuscript that addresses the points raised during the review process.

We look forward to receiving your revised manuscript.

Kind regards,

Imre Cikajlo, Ph.D.

Academic Editor

PLOS ONE

“OR received contract research fuding from FitXR https://fitxr.com/”

“OR received contract research fuding from FitXR”

4. We note that Figure 1 in your submission contain copyrighted images. All PLOS content is published under the Creative Commons Attribution License (CC BY 4.0), which means that the manuscript, images, and Supporting Information files will be freely available online, and any third party is permitted to access, download, copy, distribute, and use these materials in any way, even commercially, with proper attribution. For more information, see our copyright guidelines: http://journals.plos.org/plosone/s/licenses-and-copyright.

5. Please remove your figures from within your manuscript file, leaving only the individual TIFF/EPS image files, uploaded separately. These will be automatically included in the reviewers’ PDF.

Reviewers' comments:

Reviewer's Responses to Questions

**Comments to the Author**

1. Is the manuscript technically sound, and do the data support the conclusions?

Reviewer #1: Partly

Reviewer #2: Yes

2. Has the statistical analysis been performed appropriately and rigorously? 

Reviewer #1: No

Reviewer #2: Yes

3. Have the authors made all data underlying the findings in their manuscript fully available?

Reviewer #1: No

Reviewer #2: Yes

4. Is the manuscript presented in an intelligible fashion and written in standard English?

Reviewer #1: Yes

Reviewer #2: Yes

5. Review Comments to the Author

Reviewer #1: Plos One Review

This paper describes a study where investigators aimed to carefully match a 3D commercially available immersive exercise based application with a 2D (semi immersive) simulation. The paper is of interest to this reviewer, however methods require clarification and a more modest interpretation of the results. One concern is the claim that the gaming conditions are matched when in fact there are differences in sensory experience (haptics) and feedback that may explain some of the condition differences beyond level of immersion.

The introduction is well-written framing the study within the context of the importance of activity and the challenges to engage with it regularly. It is useful that the authors reference the COM-B model of behavior change as theoretical anchor for their study. In reviewing the references most of the citations are on healthy young persons and it might be useful to note that in the text.

A few comments on the paragraph that start with line 49-

-Is there a reference for the statement “ gaming interfaces as more accessible and affordable (line 52).” Often cost is still cited as a barrier to adoption. In fact your data indicate that as well.

-The definition of VR as 3D immersive 3D computer generated simulations (line 53-54) is too restrictive. The suggestion that 2D environments are not VR is limiting. I think there is a difference in immersion of the VR experience. If you want to be restrictive in your definition please provide a citation and rationale. I would suggest you consider immersive compared to semi-immersive VR- rather than VR and no VR.

- Can you clarify what you mean by the user can interact simulations that require full body movement (this can be achieved in 2D) and “The element of environmental interaction significantly influences…” what is that is mean by environmental interaction and how is this specific to 3D immersive environments.

A few comments on the paragraph that starts on line 78

The comments here again might best be framed when studying healthy individuals. There are studies in persons with neurologic health conditions where some of these questions have been addressed using physiologic measures and perception of effort and enjoyment.

-In the literature on VR for persons with stroke there has been work that compared commercial games (Kinect) to custom 2 games with the goal of having equivalence and addressing in this case the limitations of the commercial games.

-There has also been some work comparing the immersive to semi-immersive environment in a cycling simulation for persons with PD.

Method

It is unclear why you have selected RPE as the variable to power the study. It does not appear to be the primary question- Please provide a rationale. Also, the data from Zeng are not similar to yours as they compared their participants in VR and no-VR.

Was there any age restriction for your participants or was this a sample of convenience and participants had a narrow age range? Please state explicitly under participants.

The simulations are not matched. In addition to manipulating the level of immersion the investigators have not controlled for comparable sensory feedback (haptic) and performance feedback. Both factors likely contribute to part of the differences in the group scores. Specifically as you indicate in the discussion lines 312-318 the haptics have an enhancing effect on presence.

Please describe the four rounds of play and why they were selected or is tis this the default of the game.

Please clarify whether participants returned to their resting HR and RPE during the five-minute rest between conditions. This is important to confirm that participants were at their baseline physiology before each condition. Given that you counter-balanced the order of conditions you may analyze your resting data to confirm that they were equivalent between conditions.

Please explain why you chose the CR10 Borg instead of the original scale.

Tolerability, feasibility and acceptability typically have a priori metrics (eg. Score of X on the SSQ) is there a reason you did not chose to do this?

Please notes this as a limitation of the study. It is not clear why some of the data considered qualitative when the responses are yes and no and they are counted. Are these data just descriptive?

The analysis of your VO2 data with 27 for the 30 second epochs is interesting. How did you adjust the Bonneferone correction for that analysis? Your report analyses by round but do not describe that in the methods.

It would be better to normalize HR by age rather than present the raw values.

Was the SSQ administered at baseline and after each condition? You need a baseline assessment for it to be valid. You could analyze it with one way anova with three levels- baseline, semi and full.

A bit more explanation of the scoring on the PAAS may help the reader understand your data. Hard to know what a difference between 2.5 and 1.7 means.

Analysis-

Figure are nice

Discussion

To interpret the data it seems like you are not using a significant interaction as a measure of an effect and often interpret the main effect of condition as a finding. It would be useful to state that clearly and also to consider whether there is a meaning beyond statistics to the differences. For example, if the HR were normalized one could see if they are exercising at a meaningful intensity. Or with RPE you state that is similar but it is between 10 and 15% lower with the semi VR.

The conditions not being matched should be addressed as limitation lines 310-315.

Not stating the feasibility and acceptability a priori and measuring the SSQ at baseline are also limitations.

Finally not offering feedback in the semi VR is a limitation. Or at the very least along with the haptics speaks to have alternate explanations to immersion as solely causing the findings.

Some of your statements should be eliminated or tempered- lines 315-318 are not supported by your data with healthy participants.

Suggested specific revisions

Abstract

Line 7 Exactly matched- revised to closely matched

Reviewer #2: Peer Review for Manuscript PONE-D-24-05252: "Acute psychological and physiological benefits of exercising with virtual reality"

Major Revisions:

Introduction (Lines 25-98):

1. Detailed Mechanisms:

o Line 32-34: The introduction could benefit from a more detailed discussion on the specific mechanisms by which VR might influence physiological and psychological outcomes during exercise. This will help in setting a clear context for the study.

2. Potential Negative Effects of VR:

o Line 70-72: The introduction should include a discussion on the potential negative effects of VR exercise, such as dizziness and nausea. This is important for providing a balanced view of the technology's potential.

Methodology (Lines 99-141):

1. Sample Size Justification:

o Line 99-101: The sample size (n=17) is quite small. Power analysis details should be provided, explaining why this sample size was chosen and how it is sufficient to detect meaningful differences.

2. Participant VR Experience:

o Line 99-101: Information about participants' prior experience with VR should be included. VR's novelty could significantly influence the results, and this potential confounder should be acknowledged and controlled if possible.

3. Inclusion and Exclusion Criteria:

o Line 110-112: The criteria for including and excluding participants should be more detailed, including specific medical conditions that would disqualify participants and any other relevant factors.

4. Participant Familiarization:

o Lines 99-101: It is unclear how participants were familiarized with the VR equipment before the study. Providing details on this process is crucial for understanding the participants' comfort and proficiency with the VR system.

5. Control for Novelty Effect:

o Lines 105-108: The study should address whether any control measures were taken to account for the novelty effect of using VR, as this can significantly impact participants' responses.

Results (Lines 206-270):

1. Effect Sizes:

o Line 206-215: Effect sizes should be reported alongside p-values for all statistical tests to provide a clearer picture of the practical significance of the findings.

2. Heart Rate Discrepancy:

o Line 207-215: The heart rate results show no significant difference between VR and non-VR conditions, yet there is a main effect of time. The authors should explore and discuss possible reasons for this discrepancy.

3. Post-Hoc Analyses:

o Line 221-227: Conduct and report post-hoc analyses for VO2 and heart rate data to clarify where specific differences lie between the VR and non-VR conditions.

4. Perceived Exertion:

o Line 232-236: The perceived exertion results are higher for the VR condition, contrary to some previous findings. The discussion should include potential reasons for this difference, such as the impact of the immersive experience or differences in individual perception.

Discussion (Lines 284-362):

1. COM-B Model Discussion:

o Line 326-338: The discussion around the COM-B model appears speculative. This section should be more closely tied to the actual data from the study, providing concrete examples of how the study's findings support or challenge the COM-B model.

2. Novelty Effect:

o Line 343-344: The discussion should consider the potential impact of the novelty effect on the results and suggest ways to mitigate this in future studies.

3. Comparison with Previous Studies:

o Line 350-352: A more detailed comparison with previous studies that found lower perceived exertion in VR conditions is needed, exploring why the current study found different results.

4. Long-term Adherence:

o Line 359-360: The implications of the findings for long-term adherence to VR exercise programs should be discussed, considering the novelty might wear off over time.

5. Limitations Section:

o Add a dedicated "Limitations" subsection to discuss limitations such as the small sample size, potential novelty effects, and any other relevant factors.

Minor Revisions:

Abstract (Lines 1-19):

1. Limitations and Future Directions:

o Line 19: Include a sentence on the study's limitations and potential directions for future research to provide a more balanced overview of the study's contributions and areas for improvement.

2. Minor Grammatical Errors:

o Line 13: Minor grammatical errors and typos should be corrected for clarity.

Introduction:

1. References:

o Ensure all references are up-to-date and relevant. Some references, particularly those discussing the benefits of VR in exercise, could be expanded.

Methodology (Lines 99-141):

1. Participant Recruitment and Criteria:

o Line 105-108: Provide more detail on participant recruitment and the inclusion/exclusion criteria to enhance the study's transparency and reproducibility.

2. Experimental Setup Diagram:

o Line 127: Include a diagram of the experimental setup showing the VR and non-VR conditions to help readers visualize the experimental environment.

3. RPE Reporting in VR Condition:

o Line 154-159: Clarify how participants reported RPE in the VR condition when they could not see the scale. Describe any training or familiarization they received for this task.

Results (Lines 206-270):

1. Error Bars in Figures:

o Line 207-215: Add error bars to Figures 2 and 3 to help readers assess the variability in the data.

Discussion:

1. Minor Grammatical Corrections:

o Line 352: Minor grammatical corrections are needed for clarity.

References:

1. Formatting and Completeness:

o Line 364: Verify that all references are correctly formatted and complete, including checking DOIs and ensuring all cited works are listed in the reference section.

6. PLOS authors have the option to publish the peer review history of their article (what does this mean?). If published, this will include your full peer review and any attached files.

Reviewer #1: No

Reviewer #2: **Yes: **Cihan Aygün

---

## [Author Response · Author response to Decision Letter 0]

5 Sep 2024

These have been attached as a separate file.

---

## [Decision Letter · Decision Letter 1]

23 Oct 2024

PONE-D-24-05252R1Acute psychological and physiological benefits of exercising with virtual realityPLOS ONE

Dear Dr. Runswick,

Thank you for submitting your manuscript to PLOS ONE. After careful consideration, we feel that it has merit but does not fully meet PLOS ONE’s publication criteria as it currently stands. Therefore, we invite you to submit a revised version of the manuscript that addresses the points raised during the review process.

**ACADEMIC EDITOR: **Dear Author, in case of different comments or opinions between the reviewers, please reply to both of them in case an implementation and a possible improvement of the manuscript or part of it is possible. In case of different evaluations between the reviewers in the statistics section, please consider a possible implementation.

We look forward to receiving your revised manuscript.

Kind regards,

Angelo Rodio

Academic Editor

PLOS ONE

Reviewers' comments:

Reviewer's Responses to Questions

**Comments to the Author**

1. If the authors have adequately addressed your comments raised in a previous round of review and you feel that this manuscript is now acceptable for publication, you may indicate that here to bypass the “Comments to the Author” section, enter your conflict of interest statement in the “Confidential to Editor” section, and submit your "Accept" recommendation.

Reviewer #2: All comments have been addressed

Reviewer #3: All comments have been addressed

2. Is the manuscript technically sound, and do the data support the conclusions?

Reviewer #2: Yes

Reviewer #3: Yes

3. Has the statistical analysis been performed appropriately and rigorously? 

Reviewer #2: Yes

Reviewer #3: Yes

4. Have the authors made all data underlying the findings in their manuscript fully available?

Reviewer #2: Yes

Reviewer #3: Yes

5. Is the manuscript presented in an intelligible fashion and written in standard English?

Reviewer #2: Yes

Reviewer #3: Yes

6. Review Comments to the Author

Reviewer #2: The authors have successfully addressed my previous critiques and made the necessary revisions. The explanations regarding the statistical methods and participant selection have been clarified, and the inclusion of post-VR participant feedback has added value to the study. These revisions have strengthened the methodological rigor and reinforced the validity of the results.

Reviewer #3: Thank you for the opportunity to review this manuscript. The study addresses a highly relevant topic, investigating the psychological and physiological effects of exercise in virtual reality (VR), a field of growing interest. The use of commercially available VR devices makes the findings relevant to the general public. Among the strengths of the study, the rigorous comparison between VR exercise and a 2D screen equivalent stands out, with statistically significant results showing improvements in participants' psychological and physiological responses during the VR condition.

However, there are several sections that require adjustments to improve the clarity and coherence of the article:

Introduction: This section could be streamlined to avoid repetition of general concepts about the benefits of exercise, which are already well known. I suggest focusing more on the unique contribution of this study, namely the direct comparison between VR and 2D screen-based exercise with acute physiological and psychological measurements. This will highlight the research gap the study aims to address, avoiding unnecessary details on established knowledge.

Methodology: The description of the experimental protocol needs more clarity. It is important to specify:

How the order of conditions (VR and non-VR) was counterbalanced to avoid potential sequence effects that could influence participants' performance and psychological responses.

The exact duration of the rest period between experimental sessions should be more precisely indicated to ensure that participants adequately recovered between trials, reducing the risk of cumulative fatigue that might distort the results.

Data Analysis: A more detailed explanation of how violations of sphericity in the ANOVA were handled would be useful, along with information on any statistical adjustments made (e.g., Greenhouse-Geisser or Huynh-Feldt corrections). This would improve the transparency of the analysis and strengthen the robustness of the results.

Limitations of the study: The discussion of limitations should be expanded. The novelty of VR technology may have influenced the results, as many participants were new to this experience and may have reported higher levels of enjoyment and motivation compared to a more familiar device. This point should be mentioned, and it would be appropriate to suggest longitudinal studies to assess whether these effects persist over time once the novelty effect wears off.

Generalizability of the results: It is important to better address the fact that the participant sample consisted of young, physically active individuals, which limits the generalizability of the findings to other populations, such as older or less active individuals. I suggest including a discussion on how results might differ in these groups and proposing future research to explore these variables in different cohorts.

Conclusion section: The conclusions should be more cautious. While the results are promising, it is important not to overgeneralize. Specifically, it should be emphasized that the observed benefits may be specific to the young and healthy sample studied, and that further research is needed to confirm whether these advantages apply to other populations.

7. PLOS authors have the option to publish the peer review history of their article (what does this mean?). If published, this will include your full peer review and any attached files.

Reviewer #2: **Yes: **Cihan Aygün

Reviewer #3: **Yes: **Pierluigi Diotaiuti

---

## [Author Response · Author response to Decision Letter 1]

29 Oct 2024

Responses are in the attached response to reviewers document.

---

## [Decision Letter · Decision Letter 2]

11 Nov 2024

Acute psychological and physiological benefits of exercising with virtual reality

PONE-D-24-05252R2

Dear Dr. Runswick,

We’re pleased to inform you that your manuscript has been judged scientifically suitable for publication and will be formally accepted for publication once it meets all outstanding technical requirements.

Kind regards,

Fenghua Sun

Academic Editor

PLOS ONE

Additional Editor Comments (optional):

Reviewers' comments:

Reviewer's Responses to Questions

**Comments to the Author**

1. If the authors have adequately addressed your comments raised in a previous round of review and you feel that this manuscript is now acceptable for publication, you may indicate that here to bypass the “Comments to the Author” section, enter your conflict of interest statement in the “Confidential to Editor” section, and submit your "Accept" recommendation.

Reviewer #2: All comments have been addressed

Reviewer #3: All comments have been addressed

2. Is the manuscript technically sound, and do the data support the conclusions?

Reviewer #2: Yes

Reviewer #3: Yes

3. Has the statistical analysis been performed appropriately and rigorously? 

Reviewer #2: Yes

Reviewer #3: (No Response)

4. Have the authors made all data underlying the findings in their manuscript fully available?

Reviewer #2: Yes

Reviewer #3: Yes

5. Is the manuscript presented in an intelligible fashion and written in standard English?

Reviewer #2: Yes

Reviewer #3: Yes

6. Review Comments to the Author

Reviewer #2: The authors have successfully addressed previous critiques and made the necessary revisions. These revisions have strengthened the methodological rigor and reinforced the validity of the results.

Reviewer #3: In my assessment, the article has been meticulously reviewed and has satisfactorily met all necessary standards and criteria outlined for publication. The revisions have been thorough, addressing both content accuracy and clarity, ensuring that the work aligns with the expectations for quality and rigor. Therefore, I am confident that the article is now well-prepared to proceed to the publication stage.

7. PLOS authors have the option to publish the peer review history of their article (what does this mean?). If published, this will include your full peer review and any attached files.

Reviewer #2: **Yes: **Cihan Aygün

Reviewer #3: **Yes: **Pierluigi Diotaiuti

---

## [Editor Report · Acceptance letter]

13 Nov 2024

PONE-D-24-05252R2 

PLOS ONE

Dear Dr. Runswick, 

I'm pleased to inform you that your manuscript has been deemed suitable for publication in PLOS ONE. Congratulations! Your manuscript is now being handed over to our production team.

Kind regards, 

on behalf of

Dr. Fenghua Sun 

Academic Editor

PLOS ONE